# Towards Accelerated Autolysis? Dynamics of Phenolics, Proteins, Amino Acids and Lipids in Response to Novel Treatments and during Ageing of Sparkling Wine

**Gail B. Gnoinski [1,\*], Dugald C. Close [1], Simon A. Schmidt [2] and Fiona L. Kerslake [1]**

[1] Horticulture Centre, Tasmanian Institute of Agriculture, University of Tasmania, Sandy Bay, Hobart 7005, Australia; dugald.close@utas.edu.au (D.C.C.); fiona.kerslake@utas.edu.au (F.L.K.)

[2] The Australian Wine Research Institute, Glen Osmond, Adelaide 5064, Australia; simon.schmidt@awri.com.au

\* Correspondence: gail.gnoinski@utas.edu.au

**Abstract:** Premium sparkling wine produced by the traditional method (analogous to the French *méthode champenoise*) is characterised by the development of aged wine character as a result of a second fermentation in the bottle with lees contact and lengthy ageing. Treatments (microwave, ultrasound, or β-glucanase enzymes) were applied to disrupt the cell wall of *Saccharomyces cerevisiae* and added to the tirage liquor for the second fermentation of Chardonnay-Pinot Noir base wine *cuvée* and compared to a control, to assess effects on the release of phenolics, proteins, amino acids, and lipids at 6, 12 and 18 months post-tirage. General responses to wine ageing included a 60% increase in the total phenolic content of older sparkling wines relative to younger wines and an increase in protein concentration from 6 to 12 months bottle age. Microwave and β-glucanase enzyme treatments of yeast during tirage preparation were associated with a 10% increase in total free amino acid concentration and a 10% increase in proline concentration at 18 months bottle age, compared to control and ultrasound treatment. Furthermore, microwave treatment was associated with elevated asparagine content in wine at 18 months bottle age, relative to the control and the other wines. The β-glucanase enzyme and ultrasound treatments were associated with significant accumulation of total lipids, which were driven by 2-fold increases in the phospholipid and monoacylglycerol components in wine at 18 months bottle age and, furthermore, the microwave treatment was associated with elevated triacylglycerol at 18 months bottle age. This study demonstrates that the use of yeast treatments at the tirage stage of sparkling wine production presents an opportunity to manipulate wine composition.

**Keywords:** sparkling wine; autolysis; microwave; ultrasound; β-glucanase enzymes; lipid; protein; amino acid

## 1. Introduction

Sparkling wines produced by the traditional method follow a two-step fermentation process where grape juice is fermented into a base wine (*cuvée*) before a second step where the base wine undergoes further fermentation in a sealed bottle with a tirage liquor blend of sugar, yeast, riddling adjuvant and nutrients. The second fermentation in the bottle traps the carbon dioxide to give the characteristic effervescence of sparkling wine [1]. The yeast *Saccharomyces cerevisiae*, which is often left in a bottle following the completion of the secondary fermentation, begins to break down during prolonged exposure to lack of nutrients, high alcohol content, and carbon dioxide saturation, a process known as autolysis [2]. The ageing process is influenced by the wine's pH, temperature and duration of wine storage [3,4]. The longer that the wine is in contact with the yeast sediments (lees), the greater the complexity and changes in wine texture and flavour [1,2], an effect that motivates the extended ageing periods in premium sparkling wine production.

The slow maturation process facilitates the release of cytoplasmic (amino acids, peptides, fatty acids, nucleotides) and cell wall (glucans, mannoproteins) macromolecules from the yeast [3,5]. It is these compounds that are proposed to give rise to the so-called 'autolytic' characters of sparkling wine, often described as 'toasty', 'bread-like', or 'nutty' aromas, and 'creamy' mouthfeel [4,6]. In addition, visual attributes of sparkling wine such as foam height and stability may also be affected by variations in the concentrations of these yeast-derived compounds [7–10].

Autolysis is a process mediated by several yeast-derived enzymes. These enzymes (predominantly glucanases and proteases) break down cellular components, weaken the cell wall, and render the cell permeable, thus facilitating the release of compounds into wine [11]. Amino acids and nitrogenous components like proteins and peptides represent the major fraction of macromolecules released into wine during ageing and have been used as markers of autolysis [9,12]. Although generally in low abundance (<16 mg L$^{-1}$) [13] protein contributes to the body, sweetness, roundness, and mouthfeel of sparkling wine [14,15], facilitating the retention of aroma compounds, reducing astringency [3] and influencing foam properties in sparkling wine due to surfactant qualities [13–19].

The amino acids found in sparkling wine are initially derived from grapes, but are also released by yeast during fermentation and subsequently as a result of autolysis [20]. Furthermore, grape-derived proteins are degraded to peptides and amino acids by enzymatic processes (hydrolysis) during fermentation [20–22]. Indigenous or inoculated microflora also influence amino acid composition in wine by assimilating amino acids, along with other compounds, as nutrients for growth [23,24]. As fermentations complete, amino acid consumption gives way to passive release as organisms become quiescent and die. Moreno-Arribas et al. [25] found a positive correlation between foam height and free amino acid concentrations depending on the grape variety and sparkling wine ageing time, and Pueyo et al. [26] reported amino acids as foaming agents influencing sparkling wine foam quality. However, Puig-Deu et al. [27] found foam stability with low levels of amino acids in wine and Martinez-Lapuente et al. [28] reported that the influence of amino acids on foam parameters was insignificant.

Lipids, predominantly derived from yeast membranes, can also be released during autolysis [29,30] and are considered by some as markers of autolysis [31]. Lipid release during induced autolysis in model systems has been shown to be very rapid with sterol esters and triacylglycerols being the major lipid fractions present [29]. This rapid and early liberation of lipids is consistent with the idea that membrane disorder and structural degradation results in membrane permeability and the leakage of cytosolic constituents [2]. Temperature, pH and ethanol concentration appear to be key variables in this enzyme-mediated membrane disruption [32,33]. While the initial release of lipids into the wine matrix is rapid, their concentration tends to decrease over time due to the esterification of monoacylglycerol, diacylglycerol, triacylglycerol and fatty acids [8]. The addition of fresh lees to wine has been shown to result in a short-lived increase followed by a longer-term decrease in free fatty acids over time, as a result of lees adsorption [34,35] and residual enzymatic activity [36].

Despite the low overall concentration of lipids in sparkling wine (<2 mg L$^{-1}$), they may play important roles in the formation of volatile components, esters, ketones and aldehydes [8,37]. Furthermore, certain lipid classes have been shown to affect the bubble and foaming properties of carbonated beverages [13,31]. In beer, a positive correlation was found between free fatty acids (linolenic acid and palmitic acid) and bubble formation and size [31]. Palmitic acid has been correlated with foam height in Cava [26] with suggestions that it is the equilibrium between free and esterified forms that shape the influence of fatty acids on foamability [8]. This is consistent with the work of Culbert et al. [37] who found little effect of free fatty acids on sparkling wine foam stability.

Phenolic components have a range of effects on white wines, including influences on mouthfeel and sensitivity to oxidation [38,39]. Wine grapes are the principal source of phenolic material and wine-making practices, such as grape crushing, pressing, skin contact,

use of specific yeast strains and fining agents are the primary factors that contribute to the variation in their concentration in wine [40–42]. In addition to wine-making factors, ageing of sparkling wine on lees has also been associated with a reduction in the concentration of total phenolic material [43] and interactions between phenolic material and yeast lees have been described [44]. While the concentration of phenolic material in white sparkling wines is typically, and intentionally, low compared to other wine styles [41], they nonetheless can affect the astringency of the finished product.

Due to the length of time involved in the development of autolytic character with conventional methods, alternative approaches have been sought with the aim of introducing process efficiencies. These alternatives attempt to accelerate yeast breakdown [10,12]. Some examples of this include the use of commercial enzyme preparations, microwave heat treatment and ultrasonic disruption of yeast prior to the addition of treated yeast to the secondary fermentation. Palermo et al. [45] found β-glucanase enzyme-induced release of polysaccharides in two to three weeks in model wine relative to five months in conventional autolysis. Rodriguez-Nogales et al. [46,47] reported that the addition of β-glucanase enzymes allowed a quicker breakdown of cell walls by hydrolysis of β-glucan chains releasing mannoproteins into wine. Furthermore, the application of ultrasound and microwave energy to yeast have been shown to assist in the breakdown of yeast in a model wine laboratory system [48,49]. Garcia Martin et al. [48] found enhanced yeast lysis indicated by increased protein content in model wine. Liu et al. [49] reported that microwave treatment of lees was less effective, possibly as the treatment time was not long enough, and moderate heating was also reported to promote polysaccharide release in a model wine system. Gnoinski et al. [50] identified features consistent with cell wall disruption resulting from microwave, ultrasound or β-glucanase enzyme treatment of yeast at tirage.

The purpose of this study was to test the hypothesis that the application of microwave, ultrasound or β-glucanase enzyme treatments will promote *S. cerevisiae* yeast degradation and, when added at tirage during secondary fermentation in traditional sparkling wine production, can influence wine composition. To evaluate the effects of treatments and changes in composition associated with lees ageing, analyses of total phenolics, total proteins, amino acids and lipids were undertaken at 6, 12, and 18 months bottle age.

## 2. Materials and Methods

### 2.1. Yeast Treatments and Preparation of Sparkling Wines

To explore the effects of treatments on the wine matrix, yeast cultures ($2 \times 10^6$ cells mL$^{-1}$) were prepared from *S. cerevisiae* IOC 18-2007 (Institut Oenologique de Champagne, Lallemand, Adelaide, Australia) [51] and subjected to microwave or ultrasound or β-glucanase enzyme treatments according to the methodology described by Gnoinski et al. [50], as follows: (i) A household microwave oven (Panasonic 'the Genius' Shanghai, China, 1100 W, 50 Hz, 100% power) was used to heat 1.0 L yeast culture in a Schott bottle up to 99 °C, (ii) An ultrasonic bath (Soniclean 2000TD Ultrasonic Bath, Thebarton, Australia) was operated at 50 kHz and 350 W to treat 1.0 L yeast culture in a closed Schott bottle over five cycles of 15 min each, with circulating cold water to maintain the bath water temperature between 20 °C to 25 °C, and (iii) For the enzyme treatment, 5 g L$^{-1}$ β-glucanase enzyme (VinoTaste® Pro, Novozymes, Winequip Pty Ltd.y, Victoria, Australia) was added to 1.0 L yeast culture in a Schott bottle, mixed well, closed and maintained at 20 °C for 24 h prior to use.

To prepare the sparkling wines, a 2016 vintage base wine blend of *Vitis vinifera* L. cv. Chardonnay (63%) and Pinot Noir (37%) from sub-regions of Tasmania, Australia (pH 3.07, titratable acidity (TA) 8.63 g L$^{-1}$, alcohol content 11.26% *v/v*, free SO$_2$ 33 ppm, total SO$_2$ 133 ppm and residual sugar 0.47 g L$^{-1}$) was used. For the second fermentation in the bottle, 22.5 mL of tirage liquor was added, which comprised $2 \times 10^6$ cells mL$^{-1}$ *S. cerevisiae* IOC 18-2007 (Institut Oenologique de Champagne, Lallemand, France), 23 g L$^{-1}$ sugar, and nutrients of 0.1 g L$^{-1}$ diammonium phosphate and 0.04 g L$^{-1}$ Cerevit® (Lallemand, France) in the base wine blend media. Adjuvant, e.g., bentonite, used in riddling for sparkling wine disgorgement was excluded.

For the treatment wines, 7.5 mL of each treated yeast (microwave or ultrasound or β-glucanase enzyme treatment) was further added at a rate of 1% to each standard 750 mL bottle corresponding to that treatment, with no addition of treatments made to the controls. There were three replicates (bottles) for each treatment and the control to be sampled at each planned bottle age (6, 12, and 18 months) giving a total of 36 bottles. The bottles were capped with crown seals and positioned horizontally and stored at 15 °C in a closed, temperature-controlled refrigerator, for the secondary fermentation and aged on lees for 6, 12, or 18 months.

Disgorgement of lees from three bottles of sparkling wine of each treatment occurred after 6, 12, and 18 months. Sparkling wine bottles were riddled to a vertical position by turning twice a week for eight weeks leading up to disgorgement. After disgorgement, bottles of sparkling wine were closed with crown seals without the addition of a dosage solution and stored at 4 °C for two weeks and chemical analyses were undertaken at 6, 12, and 18 months bottle age.

### 2.2. Analysis of Basic Wine Composition

Basic enological parameters were measured according to methods previously described by Culbert et al. [37]. Samples (100 mL) in a Schott bottle were degassed prior to chemical analyses using an ultrasonic bath (Sonorex Digitec, Bandelin Electronic GmbH & Co., Berlin, Germany) for 10 min. Titratable acidity (TA, expressed as g $L^{-1}$ tartaric acid) and pH were determined by auto-titrator (Compact Titrator, Crison Instruments SA, Allela, Spain). Alcohol content (%) was determined by alcolyser (Anton Paar, Graz, Austria) and residual sugar (glucose and fructose) was measured by enzymatic kit (Boehringer-Mannheim, R-BioPharm, Darmstadt, Germany) and spectrophotometric reading (Infinite M200 Pro, Groedig, Austria), at 6 months bottle age only. Total phenolics were determined from the absorbance (280 nm) of degassed wine using a UV-Vis spectrophotometer (GBC Scientific Equipment, Melbourne, Australia) [37]. In-bottle analyses of sparkling wine tirage fermentation and maturation at 3, 6, and 12 months bottle age were undertaken by non-destructive NIR technology, the Bevscan (Jeffress Engineering, Adelaide, Australia), designed for screening wine quality [52,53].

### 2.3. Analysis of Proteins

Total protein concentrations in the sparkling wines were analysed using the method published by Culbert et al. [37]. Haze-forming proteins, including chitinase and thaumatin-like proteins, were measured for all the triplicates for the treatments and control. Samples of degassed wines (1.5 mL) were filtered (0.45 micron, PVDF syringe filter, Merck Millipore, MA, USA) and sterilised prior to injection on an Agilent 1260 (Palo Alto, CA, USA) ultra-high performance liquid chromatography (UHPLC) equipped with a Prozap C18 column (10 × 2.1 mm, Agilent, Palo Alto, CA, USA). Separation was achieved with two solvents, solvent A was 1% TFA/$H_2O$ and solvent B was 0.1% TFA/ACN, at a flow rate of 0.75 mL per minute and a mobile gradient of 0 to 1 min (10 to 20% solvent B), 1 to 4 min (20 to 40% solvent B), 4 to 6 min (40 to 80% solvent B), 6 to 7 min (80% solvent B) and 7 to 10 min (1% solvent B). Proteins were detected at 210 nm by an Agilent UV/Vis detector and identified by retention times and comparison with isolation standards and quantified (as mg $L^{-1}$ thaumatin equivalents) on calibration curves.

### 2.4. Analysis of Amino Acids

The quantitation of amino acids was performed using a derivatisation technique with 6-aminoquinolyl-N-hydroxysuccinimidyl carbamate (AQC), according to procedures previously published by Culbert et al. [37]. Analyses were performed on the Agilent 1290 Infinity liquid chromatography system coupled with Agilent 6490 Triple Quadrupole mass spectrometry (LC-MS) with iFunnel technology (in ESI negative and positive mode). Samples were diluted (1:100) with 0.1% formic acid in Milli-Q-Water to an appropriate concentration. Calibration curves were acquired in the range of 0.05 μmol $L^{-1}$ per 100 μmol $L^{-1}$. A 2 mL

eppendorf tube was filled with 10 μL of sample/calibrant and 70 μL of borate buffer (borate 200 mmol $L^{-1}$, tris-(2-carboxyethyl)-phosphine 1 mmol $L^{-1}$, Ascorbate 1 mmol $L^{-1}$), followed by vortexing (10 s) and centrifugation twice with the addition of 20 μL of AQC in between). The solution was then heated for 10 min at 55 °C, centrifuged at 4 °C for 10 min, and transferred to HPLC vials with insert for chemical analysis by LC-MS/MS ESI (positive) mode.

The high-performance liquid chromatography (HPLC) conditions were as follows: Injection volume was 1 μL, the flow rate was 0.8 mL per minute, Solvent A was 0.1% formic acid in Milli-Q-Water, Solvent B was 0.1% formic acid in Acetonitrile, using an Agilent Zorbax Eclipse Plus C18 RRHD column (2.1 × 100 mm 1.8-micron) and column temperature of 60 °C. The HPLC mobile gradient was as follows: 0 to 0.5 min (1% solvent B), 3.5 min (10% solvent B), 6.0 min (15% solvent B), 6.5 min (20% solvent B), 6.6 min (75% solvent B), 7.5 min (75% solvent B), 7.6 to 10 min (1% solvent B). The mass spectrometry parameters were set at a gas temperature of 315 °C, the gas flow was set at 14 L per minute, the nebuliser pressure was set at 40 psi, sheath gas flow was set at 3800 V, nozzle voltage was set at 1500 V and a start time at 0 min. Calibration curves and amino acid concentrations were calculated using Agilent MassHunter (v A.00.06.36) software.

## 2.5. Analysis of Lipids

Sample preparation for the analysis of lipids in sparkling wines involved a liquid-liquid extraction on a concentrated wine sample using a modified protocol from Bligh and Dryer [54]. A filtered (0.45 μm) wine sample (300 mL) was concentrated to 10 mL using rotary evaporation. The concentrated wine sample was extracted with DCM (1 × 20 mL; 1 × 10 mL, 1 × 5 mL) and the DCM phases (lower layer) combined into a centrifuge tube and centrifuged (2 min, 3200 rpm). The layer containing the lipid was separated and evaporated in a Turbo Vac tube at 42 °C for 20 min to evaporate the DCM down to 1 mL. The total solvent extract (lipid-bearing layer) was transferred to a pre-weighed GC vial, using three washes in the Turbo Vac tube. The total solvent extract was dried down in a vial and weighed to determine the weight of the total lipid, and resuspended in 1 mL DCM for Iatroscan lipid class analysis. Equipment was thoroughly cleaned between samples, to avoid cross-contamination. Samples were stored at −80 °C before further analyses.

An aliquot of the total solvent extract was analysed using an Iatroscan MK VI TH10 thin-layer chromatography-flame ionisation detector (TLC-FID) analyser (Japan, Tokyo) to quantify individual lipid classes [55,56]. Samples were applied in duplicate to silica gel SIII chromarods (5 μm particle size) using 1 μL micropipettes and allowed to develop. A solvent system with mobile phase, hexane-diethyl ether formic acids (60:15:1.5 *v/v/v*), was used to separate non-polar lipids, e.g., monoacylglycerol (MAG), diacylglycerol (DAG), triacylglycerol (TAG), free fatty acids (FFA), and sterol esters (SE). The chromarods were oven dried following development and subsequently analysed. Calibration of the flame ionisation detector was undertaken for each lipid class using monopalmitin (MAG), dipalmitin (DAG), tripalmitin (TAG), palmitic acid (FFA), cholesteryl palmitate (SE), and phosphatidylcholine (PL). Peaks were quantified on a Windows 10 compatible computer using SIC u7 Data Station for Iatroscan software (LSI Medience Corporation, System Instruments CO. Ltd., Tokyo, Japan). Lipid class peaks were labelled using SIC u7 Iatroscan Integrating Software V2.1, quantified using predetermined linear regressions, and expressed as per sample. Total lipid amounts for each sample were determined as the sum of all lipid classes for each sample.

## 2.6. Statistical Analyses

One-way analysis of variance (ANOVA) with post hoc Dunnett's multiple comparison's test at $p \leq 0.05$ were performed using GraphPad Prism version 8 for MAC (GraphPad Software, San Diego, CA, USA). One-way ANOVA was considered significant where $p \leq 0.05$, and interactions and main effects between yeast treatments and bottle age time points were identified using two-way ANOVA with post hoc Tukey's Test at $p \leq 0.05$.

## 3. Results

### 3.1. Basic Composition of Sparkling Wines

The base wine cuvée of Vitis vinifera L. cv. Chardonnay (63%) and Pinot Noir (37%) had a pre-bottling composition pH of 3.07, titratable acidity (TA) of 8.63 g L$^{-1}$, the alcohol content of 11.26% (*v/v*), and residual sugar of 0.47 g L$^{-1}$. There were differences between treatments but only at 18 months bottle age. After treatment, the wines were of similar pH and titratable acidity levels, respectively, at 6 and 12 months bottle age (Table 1). At 18 months, however, there were significantly lower pH ($p$ = 0.022) and TA ($p$ = 0.028) in treated wines as compared to the control. While the effects were statistically significant between treatments at 18 months, the differences in values would not be considered to result in perceived sensory differences in wine (i.e., pH 3.06 to 3.09; TA 8.1 and 8.2 g L$^{-1}$). There was no evidence that residual sugar (mean = 1.13 g L$^{-1}$) or alcohol concentrations (13% *v/v*) differed according to treatment at 6 months (Table 1).

**Table 1.** Composition [1] of sparkling wines by treatment after 6, 12 and 18 months bottle ageing.

| Analyte | Bottle Age (Months) | Control | Microwave | Ultrasound | β-Glucanase Enzyme | $p$ Value (Treatment) |
|---|---|---|---|---|---|---|
| Alcohol (% *v/v*) | 6 | 12.9 ± 0.01 | 13.1 ± 0.5 | 13.0 ± 0.2 | 12.9 ± 0.03 | 0.800 |
| Residual sugar (mg L$^{-1}$) | 6 | 1.18 ± 0.01 | 1.09 ± 0.14 | 1.14 ± 0.05 | 1.09 ± 0.17 | 0.730 |
| pH | 6 | 3.09 ± 0.00 | 3.09 ± 0.01 | 3.09 ± 0.00 | 3.09 ± 0.01 | 0.330 |
|  | 12 | 3.12 ± 0.01 | 3.12 ± 0.01 | 3.12 ± 0.01 | 3.11 ± 0.01 | 0.399 |
|  | 18 | 3.07 ± 0.00 [a] | 3.06 ± 0.00 [b] | 3.08 ± 0.02 [a] | 3.09 ± 0.01 [a] | 0.022 |
| $p$ value (time) | <0.0001 | | | | | |
| TA (g L$^{-1}$) | 6 | 8.3 ± 0.1 | 8.3 ± 0.1 | 8.2 ± 0.1 | 8.2 ± 0.1 | 0.611 |
|  | 12 | 8.0 ± 0.0 | 8.0 ± 0.1 | 8.0 ± 0.0 | 8.0 ± 0.1 | 0.330 |
|  | 18 | 8.2 ± 0.0 [a] | 8.1 ± 0.1 [b] | 8.1 ± 0.1 [b] | 8.1 ± 0.1 [b] | 0.028 |
| $p$ value (time) | <0.0001 | | | | | |
| Total phenolics (au) | 6 | 0.89 ± 0.03 | 0.87 ± 0.02 | 0.89 ± 0.03 | 0.87 ± 0.01 | 0.373 |
|  | 12 | 1.17 ± 0.04 | 1.16 ± 0.02 | 1.15 ± 0.01 | 1.16 ± 0.02 | 0.732 |
|  | 18 | 1.44 ± 0.02 | 1.38 ± 0.05 | 1.40 ± 0.04 | 1.41 ± 0.01 | 0.222 |
| $p$ value (time) | <0.0001 | | | | | |
| Total proteins (mg L$^{-1}$) | 6 | 32.9 ± 1.1 | 32.4 ± 2.4 | 31.3 ± 1.7 | 32.6 ± 1.3 | 0.686 |
|  | 12 | 87.5 ± 4.7 | 80.4 ± 14.9 | 83.6 ± 1.2 | 69.0 ± 10.1 | 0.167 |
|  | 18 | 75.6 ± 0.5 | 75.5 ± 4.1 | 78.9 ± 1.9 | 77.4 ± 0.5 | 0.096 |
| $p$ value (time) | <0.0001 | | | | | |
| Total free amino acids (mg L$^{-1}$) | 6 | 707 ± 40 | 659 ± 65 | 695 ± 17 | 686 ± 13 | 0.780 |
|  | 12 | 634 ± 27 | 633 ± 11 | 619 ± 19 | 622 ± 27 | 0.783 |
|  | 18 | 616 ± 33 [a] | 686 ± 20 [a] | 628 ± 28 [a] | 694 ± 44 [b] | 0.044 |
| $p$ value (time) | 0.005 | | | | | |

[1] Data are mean ± SD, *n* = 3, one-way ANOVA significance ($p$ ≤ 0.05.) conducted among yeast treatments compared to the control, and superscript letters indicate significantly different values according to Dunnett's multiple comparison's test where applicable. Different letters (within rows) indicate statistical significance (comparing across treatments, not timepoints). Main effects of treatment and time with their interactions determined by two-way ANOVA with post hoc Tukey's Test ($p$ ≤ 0.05). Titratable acidity (TA) expressed as g of tartaric acid L$^{-1}$ and total protein measured as mg L$^{-1}$ thaumatin.

### 3.2. Total Phenolics of Sparkling Wines

The total concentration of phenolic substances increased with bottle age ($p$ < 0.0001) from a mean of 0.88 au at six months (across all treatments) to 1.41 au at 18 months (Table 1). Although the concentration of phenolic substances changed with age there was no evidence that their concentrations were affected by yeast treatment at any time point.

### 3.3. Total Protein of Sparkling Wines

There were no interactive effects of yeast treatments and bottle age on protein concentrations. Total protein concentrations were similar between the control and the treated wines with an average of 32 mg L$^{-1}$ at six months bottle age (Table 1). Between 6 and

12 months, the total protein concentrations increased across all treatments to an average of 80 mg $L^{-1}$ ($p < 0.0001$), and did not increase further between 12 and 18 months (76.8 mg $L^{-1}$).

### 3.4. Free Amino Acid Content of Sparkling Wines

There were interactive effects of yeast treatments and bottle age on free amino acid concentrations. During the early phases of bottle ageing, there was no evidence for an effect of yeast treatment on free amino acid concentration which averaged 690 and 627 mg $L^{-1}$ at 6 and 12 months respectively (Table 1). However, at 18 months bottle age there was evidence that the β-glucanase enzyme-treated wine had a higher total free amino acid concentration than both the control (mean difference = 78 mg $L^{-1}$, $p = 0.044$), and the ultrasound treated wines (mean difference = 66 mg $L^{-1}$) (Table 1).

Overall, the total free amino acids concentration decreased in the control wine between 6 and 18 months bottle age (mean decrease = 91 mg $L^{-1}$, $p = 0.023$). As with the control wines, the concentrations of free amino acids in the microwave treated wines appeared to decrease between 6 and 12 months (mean decrease = 26 mg $L^{-1}$) but this was not supported by analysis ($p = 0.832$). From 12 to 18 months, free amino acid concentrations increased (mean increase = 53 mg $L^{-1}$, $p = 0.032$) with no evidence for a net change over 6 to 18 months. Similarly, the β-glucanase enzyme treatment wine had a decrease in total free amino acids between 6 and 12 months bottle age ($p = 0.043$), which then recovered to a concentration that was similar to what it was at 6 months ($p = 0.957$). The concentrations of free amino acids in the ultrasound treatment wines appeared to decrease between 6 and 18 months (mean decrease = 67 mg $L^{-1}$) with no evidence for a change over 12 and 18 months ($p = 0.384$).

At all bottle ages, proline was the predominant amino acid accounting for 64% of the total free amino acid, followed by arginine (7%), asparagine (5%), and lysine (4%) (Table 2). Glutamic acid, alanine, and aspartic acid constituted about 3% each of the total free amino acid, while glycine, leucine, and phenylalanine formed about 2% of each of the total amino acids. Tyrosine, valine, serine, threonine, cysteine, isoleucine, methionine, and tryptophan were present in minor abundances of about 1% each. Glutamine and histidine were below the levels of analytical detection.

**Table 2.** Free amino acid concentrations [1] (mg $L^{-1}$) of sparkling wines after 6, 12 and 18 months ageing.

| Analyte | Bottle Age (Months) | Treatments | | | | |
|---|---|---|---|---|---|---|
| | | Control | Microwave | Ultrasound | β-Glucanase Enzyme | *p* Value (Treatment) |
| Proline | 6 | 454.0 ± 26.2 | 417.0 ± 42.6 | 447.0 ± 15.9 | 437.0 ± 3.91 | 0.411 |
| | 12 | 402 ± 14.0 | 405.0 ± 6.26 | 396.0 ± 20.4 | 394.0 ± 15.1 | 0.758 |
| | 18 | 412.0 ± 22.2 [a] | 464.0 ± 17.4 [b] | 421.0 ± 14.6 [a] | 461.0 ± 25.7 [b] | 0.028 |
| *p* value (time) | 0.002 | | | | | |
| Arginine | 6 | 51.1 ± 3.16 | 47.8 ± 6.45 | 47.8 ± 0.31 | 47.6 ± 4.83 | 0.721 |
| | 12 | 46.1 ± 3.83 | 46.5 ± 2.22 | 46.3 ± 3.16 | 47.8 ± 1.99 | 0.900 |
| | 18 | 39.9 ± 3.42 | 40.7 ± 0.90 | 38.0 ± 2.93 | 44.5 ± 2.47 | 0.079 |
| *p* value (time) | 0.0005 | | | | | |
| Asparagine | 6 | 36.1 ± 3.21 | 35.4 ± 3.29 | 35.7 ± 2.11 | 34.2 ± 0.79 | 0.825 |
| | 12 | 22.6 ± 0.38 | 23.7 ± 1.01 | 23.9 ± 0.76 | 23.5 ± 2.36 | 0.692 |
| | 18 | 15.2 ± 0.78 [a] | 17.6 ± 0.50 [b] | 16.5 ± 1.06 [a] | 16.9 ± 0.56 [a] | 0.012 |
| *p* value (time) | <0.0001 | | | | | |
| Lysine | 6 | 25.4 ± 1.32 | 24.8 ± 1.35 | 25.6 ± 0.50 | 25.4 ± 0.70 | 0.770 |
| | 12 | 24.3 ± 0.72 | 24.5 ± 0.32 | 23.0 ± 0.29 | 23.6 ± 1.34 | 0.140 |
| | 18 | 23.1 ± 0.97 | 25.4 ± 1.50 | 22.8 ± 1.63 | 24.7 ± 2.04 | 0.219 |
| *p* value (time) | 0.032 | | | | | |
| Glutamic acid | 6 | 23.5 ± 2.00 | 21.8 ± 1.49 | 23.0 ± 0.36 | 22.8 ± 1.12 | 0.540 |
| | 12 | 21.4 ± 1.71 | 21.2 ± 0.35 | 20.1 ± 0.70 | 21.5 ± 1.65 | 0.540 |
| | 18 | 19.1 ± 1.62 | 21.4 ± 0.42 | 19.8 ± 1.44 | 21.9 ± 1.18 | 0.082 |
| *p* value (time) | <0.0001 | | | | | |

**Table 2.** *Cont.*

| Analyte | Bottle Age (Months) | Treatments | | | | |
|---|---|---|---|---|---|---|
| | | Control | Microwave | Ultrasound | β-Glucanase Enzyme | *p* Value (Treatment) |
| Alanine | 6 | 23.1 ± 2.08 | 21.3 ± 2.02 | 22.3 ± 0.21 | 22.8 ± 1.34 | 0.580 |
| | 12 | 22.0 ± 1.81 | 21.3 ± 0.36 | 20.4 ± 0.31 | 21.1 ± 1.57 | 0.500 |
| | 18 | 20.8 ± 0.72 | 22.1 ± 0.14 | 20.9 ± 2.33 | 23.6 ± 1.42 | 0.129 |
| *p* value (time) | 0.040 | | | | | |
| Aspartic acid | 6 | 18.2 ± 1.27 | 17.7 ± 1.50 | 17.6 ± 0.92 | 18.7 ± 1.22 | 0.710 |
| | 12 | 16.2 ± 1.30 | 16.5 ± 0.21 | 15.5 ± 0.30 | 15.3 ± 0.75 | 0.280 |
| | 18 | 14.7 ± 0.95 | 16.4 ± 0.86 | 15.0 ± 0.93 | 17.0 ± 2.11 | 0.175 |
| *p* value (time) | 0.0004 | | | | | |
| Leucine | 6 | 13.2 ± 0.56 | 12.2 ± 1.14 | 13.3 ± 0.10 | 14.2 ± 1.73 | 0.240 |
| | 12 | 17.6 ± 2.11 | 15.9 ± 0.40 | 15.6 ± 0.15 | 15.8 ± 0.56 | 0.180 |
| | 18 | 15.5 ± 1.91 | 16.4 ± 0.45 | 15.5 ± 0.85 | 16.8 ± 0.91 | 0.448 |
| *p* value (time) | <0.0001 | | | | | |
| Glycine | 6 | 13.0 ± 0.81 | 12.4 ± 1.31 | 12.7 ± 0.36 | 12.9 ± 1.10 | 0.870 |
| | 12 | 11.1 ± 0.79 | 10.2 ± 0.23 | 10.2 ± 0.30 | 10.3 ± 0.61 | 0.190 |
| | 18 | 9.59 ± 1.17 | 10.8 ± 0.43 | 9.96 ± 0.80 | 11.7 ± 2.68 | 0.378 |
| *p* value (time) | 0.001 | | | | | |
| Phenylalanine | 6 | 12.1 ± 0.50 | 11.5 ± 0.69 | 12.0 ± 0.36 | 12.5 ± 0.75 | 0.280 |
| | 12 | 13.1 ± 1.27 | 12.0 ± 0.59 | 12.3 ± 0.46 | 12.6 ± 1.21 | 0.520 |
| | 18 | 11.5 ± 0.95 | 12.3 ± 0.44 | 11.7 ± 0.63 | 12.8 ± 0.67 | 0.157 |
| *p* value (time) | 0.331 | | | | | |
| Tyrosine | 6 | 9.0 ± 0.62 | 8.77 ± 0.81 | 9.27 ± 0.06 | 9.63 ± 0.12 | 0.270 |
| | 12 | 9.30 ± 0.72 | 8.67 ± 0.58 | 9.00 ± 0.62 | 9.33 ± 1.02 | 0.690 |
| | 18 | 8.03 ± 0.65 | 8.13 ± 0.25 | 8.33 ± 0.40 | 9.57 ± 0.99 | 0.063 |
| *p* value (time) | 0.052 | | | | | |
| Valine | 6 | 6.9 ± 0.20 | 6.6 ± 0.76 | 6.83 ± 0.21 | 6.97 ± 0.06 | 0.710 |
| | 12 | 6.67 ± 0.38 | 6.63 ± 0.15 | 6.50 ± 0.10 | 6.57 ± 0.12 | 0.790 |
| | 18 | 6.29 ± 0.54 | 6.92 ± 0.35 | 6.40 ± 0.31 | 7.34 ± 0.53 | 0.065 |
| *p* value (time) | 0.037 | | | | | |
| Serine | 6 | 5.67 ± 0.59 | 5.93 ± 1.14 | 5.67 ± 0.35 | 5.50 ± 0.44 | 0.900 |
| | 12 | 5.83 ± 0.45 | 5.50 ± 0.27 | 5.63 ± 0.15 | 5.60 ± 0.44 | 0.700 |
| | 18 | 4.97 ± 0.18 | 5.65 ± 0.22 | 5.16 ± 0.54 | 5.28 ± 0.09 | 0.162 |
| *p* value (time) | 0.129 | | | | | |
| Threonine | 6 | 5.03 ± 0.55 | 5.13 ± 0.55 | 5.23 ± 0.15 | 5.17 ± 0.40 | 0.960 |
| | 12 | 4.70 ± 0.27 | 4.70 ± 0.20 | 4.57 ± 0.15 | 4.67 ± 0.25 | 0.860 |
| | 18 | 4.57 ± 0.32 | 5.19 ± 0.09 | 4.83 ± 0.27 | 5.36 ± 1.58 | 0.652 |
| *p* value (time) | 0.160 | | | | | |
| Cysteine | 6 | 3.60 ± 0.27 | 3.27 ± 0.21 | 3.43 ± 0.06 | 3.40 ± 0.17 | 0.280 |
| | 12 | 4.17 ± 0.40 | 3.97 ± 0.45 | 4.00 ± 0.53 | 4.17 ± 0.51 | 0.930 |
| | 18 | 4.50 ± 0.36 | 5.14 ± 0.72 | 5.09 ± 0.19 | 5.12 ± 0.49 | 0.352 |
| *p* value (time) | <0.0001 | | | | | |
| Isoleucine | 6 | 3.50 ± 0.10 | 3.47 ± 0.23 | 3.60 ± 0.10 | 3.70 ± 0.20 | 0.380 |
| | 12 | 4.30 ± 0.30 | 4.07 ± 0.15 | 3.97 ± 0.06 | 4.00 ± 0.20 | 0.240 |
| | 18 | 3.73 ± 0.32 | 4.04 ± 0.13 | 3.80 ± 0.21 | 4.19 ± 0.46 | 0.284 |
| *p* value (time) | 0.001 | | | | | |
| Methionine | 6 | 2.97 ± 0.25 | 2.97 ± 0.25 | 3.03 ± 0.06 | 2.97 ± 0.12 | 0.960 |
| | 12 | 2.80 ± 0.26 | 2.63 ± 0.15 | 2.67 ± 0.06 | 2.67 ± 0.21 | 0.713 |
| | 18 | 2.85 ± 0.13 | 3.11 ± 0.02 | 3.00 ± 0.26 | 3.09 ± 0.18 | 0.277 |
| *p* value (time) | 0.002 | | | | | |
| Tryptophan | 6 | 0.40 ± 0.00 | 0.43 ± 0.06 | 0.43 ± 0.06 | 0.40 ± 0.00 | 0.600 |
| | 12, 18 | bd | bd | bd | bd | - |
| Histidine | 6, 12, 18 | bd | bd | bd | bd | - |
| Glutamine | 6, 12, 18 | bd | bd | bd | bd | - |

[1] Data are mean ± SD, *n* = 3, one-way ANOVA significance ($p \leq 0.05$.) conducted among yeast treatments compared to the control, and superscript letters indicate significantly different values according to Dunnett's multiple comparison's test where applicable. Different letters (within rows) indicate statistical significance (comparing across treatments, not timepoints). Main effects of treatment and time with their interactions determined by two-way ANOVA with post hoc Tukey's Test ($p \leq 0.05$). Analyses below detection limit are denoted as "bd".

For total free, or individual, amino acids there were differences between treatments but only at 18 months bottle age whereby the β-glucanase enzyme and microwave treated wines had elevated proline levels ($p = 0.028$) compared to the control and the ultrasound treated wines (Table 2). Asparagine was also elevated ($p = 0.012$) by 8% in the microwave treated wine, relative to the other wines. Other than asparagine and proline there was no evidence for other differences in amino acid composition at 18 months bottle age.

Over time, the concentrations of proline, arginine, asparagine, lysine, glutamic acid, alanine, aspartic acid, glycine, tyrosine, valine, and methionine decreased, whereas leucine, cysteine, and isoleucine concentrations increased, with bottle age (Table 2).

### 3.5. Lipid Compositon of Sparkling Wines

The influence of yeast treatments and wine ageing on the lipid composition (i.e. total lipids, free fatty acids, phospholipids, monoacylglycerol, diacylglycerol, triacylglycerol, and sterol esters) of the sparkling wines was investigated (Table 3). There were significant interaction effects of yeast treatment and bottle age on total lipids ($p = 0.0005$), phospholipids ($p = 0.018$), DAG ($p = 0.017$) and MAG ($p = 0.010$).

**Table 3.** Lipid composition [1] ($\mu g\ mL^{-1}$) of sparkling wines after 6, 12 and 18 months ageing.

| Analyte | Bottle Age (Months) | Control | Microwave | Ultrasound | β-Glucanase Enzyme | *p* Value (Treatment) |
|---|---|---|---|---|---|---|
| Total lipid | 6 | $6.26 \pm 1.32$ | $6.94 \pm 2.36$ | $5.56 \pm 0.96$ | $4.36 \pm 0.62$ | 0.127 |
| | 12 | $3.95 \pm 0.50$ [a] | $4.76 \pm 0.29$ [a] | $3.51 \pm 0.45$ [a] | $5.58 \pm 1.16$ [b] | 0.048 |
| | 18 | $4.13 \pm 1.02$ [a] | $4.41 \pm 0.33$ [a] | $5.42 \pm 0.24$ [a] | $6.61 \pm 1.10$ [b] | 0.013 |
| *p* value (time) | 0.089 | | | | | |
| Free fatty acids (FFA) | 6 | $0.46 \pm 0.10$ | $0.45 \pm 0.11$ | $0.41 \pm 0.08$ | $0.38 \pm 0.08$ | 0.647 |
| | 12 | $0.13 \pm 0.03$ | $0.13 \pm 0.01$ | $0.08 \pm 0.03$ | $0.14 \pm 0.05$ | 0.173 |
| | 18 | $0.10 \pm 0.02$ [a] | $0.08 \pm 0.06$ [a] | $0.07 \pm 0.02$ [a] | $0.19 \pm 0.02$ [b] | 0.046 |
| *p* value (time) | <0.0001 | | | | | |
| Phospholipids (PL) | 6 | $0.94 \pm 0.13$ | $2.08 \pm 1.78$ | $0.98 \pm 0.19$ | $0.73 \pm 0.03$ | 0.165 |
| | 12 | $0.57 \pm 0.13$ [a] | $0.62 \pm 0.35$ [a] | $0.32 \pm 0.09$ [a] | $1.04 \pm 0.10$ [b] | 0.014 |
| | 18 | $0.47 \pm 0.19$ [a] | $0.42 \pm 0.08$ [a] | $1.10 \pm 0.16$ [b] | $1.14 \pm 0.30$ [b] | 0.005 |
| *p* value (time) | | | | | | 0.059 |
| Diacylglycerol (DAG) | 6 | $1.34 \pm 0.78$ | $1.06 \pm 0.49$ | $0.87 \pm 0.31$ | $0.75 \pm 0.26$ | 0.421 |
| | 12 | $2.30 \pm 0.23$ [a] | $2.78 \pm 0.15$ [a] | $2.07 \pm 0.18$ [a] | $3.13 \pm 0.74$ [b] | 0.049 |
| | 18 | $2.56 \pm 0.71$ | $2.61 \pm 0.30$ | $2.86 \pm 0.23$ | $3.78 \pm 0.58$ | 0.055 |
| *p* value (time) | <0.0001 | | | | | |
| Triacylglycerol (TAG) | 6 | $0.04 \pm 0.03$ | $0.07 \pm 0.07$ | $0.04 \pm 0.02$ | $0.03 \pm 0.02$ | 0.585 |
| | 12 | $0.05 \pm 0.03$ | $0.06 \pm 0.02$ | $0.05 \pm 0.04$ | $0.07 \pm 0.06$ | 0.760 |
| | 18 | $0.02 \pm 0.01$ [a] | $0.09 \pm 0.01$ [b] | $0.04 \pm 0.05$ [a] | $0.07 \pm 0.01$ [a] | 0.048 |
| *p* value (time) | 0.516 | | | | | |
| Monoacylglycerol (MAG) | 6 | $3.08 \pm 0.48$ | $2.94 \pm 0.38$ | $2.87 \pm 0.57$ | $2.22 \pm 0.44$ | 0.115 |
| | 12 | $0.54 \pm 0.08$ | $0.67 \pm 0.08$ | $0.65 \pm 0.21$ | $0.91 \pm 0.25$ | 0.144 |
| | 18 | $0.65 \pm 0.11$ [a] | $0.76 \pm 0.08$ [a] | $1.04 \pm 0.07$ [b] | $1.07 \pm 0.16$ [b] | 0.0003 |
| *p* value (time) | <0.0001 | | | | | |
| Sterol esters (SE) | 6 | $0.41 \pm 0.21$ | $0.32 \pm 0.26$ | $0.40 \pm 0.21$ | $0.25 \pm 0.10$ | 0.661 |
| | 12 | $0.36 \pm 0.10$ [a] | $0.51 \pm 0.05$ [b] | $0.35 \pm 0.04$ [a] | $0.29 \pm 0.08$ [a] | 0.034 |
| | 18 | $0.32 \pm 0.01$ | $0.46 \pm 0.11$ | $0.30 \pm 0.04$ | $0.37 \pm 0.06$ | 0.202 |
| *p* value (time) | 0.516 | | | | | |

[1] Data are mean ± SD, *n* = 3, one-way ANOVA significance ($p \le 0.05$.) conducted among yeast treatments compared to the control, and superscript letters indicate significantly different values according to Dunnett's multiple comparison's test where applicable. Different letters (within rows) indicate statistical significance (comparing across treatments, not timepoints). Main effects of treatment and time with their interactions determined by two-way ANOVA with post hoc Tukey's Test ($p \le 0.05$).

Total lipid concentration was similar across all treatments, ranging between 4.36 to 6.94 $\mu g\ mL^{-1}$ after six months bottle age (Table 3). At 12 months bottle age, the β-glucanase enzyme treatment wine had a higher total lipid concentration (5.58 $\mu g\ mL^{-1}$) compared to the other treatments (range 3.51 to 4.76 $\mu g\ mL^{-1}$) (Table 3) that was associated with elevated concentrations of phospholipids (1.04 $\mu g\ mL^{-1}$) and diacylglycerol

(3.13 µg mL$^{-1}$). Furthermore, at 12 months bottle age the microwave treatment wines had elevated concentrations of sterol esters (0.51 µg mL$^{-1}$) relative to the other treatments.

At 18 months bottle age, the β-glucanase enzyme treatment wine had a higher total lipid concentration (6.61 µg mL$^{-1}$) compared to the other treatments (range from 4.13 to 5.42 µg mL$^{-1}$) (Table 3), associated with elevated concentrations of free fatty acids (0.19 µg mL$^{-1}$), phospholipids (1.14 µg mL$^{-1}$), triacylglycerol (0.07 µg mL$^{-1}$) and monoacylglycerol (1.07 µg mL$^{-1}$). Furthermore, at 18 months bottle age the ultrasound treatment had elevated phospholipid (1.10 µg mL$^{-1}$) and monoacylglycerol (1.04 µg mL$^{-1}$) concentrations, relative to other treatments, but similar to the β-glucanase enzyme-treated wine, and the microwave treatment had elevated triacylglycerol (0.09 µg mL$^{-1}$) concentrations.

The proportions of the phospholipids were over 2-fold higher in the ultrasound treatment wine (forming 20% of the total lipid content) and β-glucanase enzyme treatment wine (forming 17% of the total lipid content) compared to the rest of the wines (forming 10% of the total lipid content), at 18 months bottle age (Table 3).

Overall, total lipid concentrations were significantly different between 6 months and 12 months bottle age ($p = 0.016$) in the ultrasound-treated wine. Furthermore, at age 18 months there were significant increases in total lipid concentrations in the ultrasound treated wine ($p = 0.028$) compared to values at 12 months bottle age, and the β-glucanase enzyme ($p = 0.019$) treated wine compared to values at 6 months bottle age.

## 4. Discussion

This study investigated changes to the compositional matrix (total phenolic material, total protein, amino acid, and lipid concentrations) in wine following the addition of yeast subjected to lysis treatments to the tirage solution during the second fermentation step of the traditional method sparkling wine production. The objective was to explore the effects of the microwave, ultrasound or β-glucanase enzyme treatments that were designed to disrupt yeast cell walls and hasten autolysis over 6, 12 and 18 months bottle ageing. Increases in total free amino acid and total lipid concentrations were the major compositional effects observed in response to these treatments. Other parameters, such as the concentrations of total protein and phenolic material, changed over time but not in response to yeast treatment. These compositional changes are discussed in more detail below.

At 18 months of ageing, the total free amino acid concentration was 10% higher in the wines to which microwave and β-glucanase enzyme-treated yeasts had been added at tirage relative to either the control or wines that received ultrasound treated yeast. These differences in total amino acids were largely driven by increases in proline concentration that accounted for 67% of the total amino acid pool and increased by 46 g L$^{-1}$ in microwave and enzyme treatments. These differences were not evident until 18 months of bottle storage and reflect both decreases in proline concentration in control and ultrasound treated wines and increases in microwave and enzyme-treated wines relative to their respective concentrations at six months. This was possibly related to treatment-induced stress on the yeast that triggered cell degradation [4,9] and autolysis associated with long-term cellular degradation previously reported by Gnoinski et al. [50] who found that yeast cell walls were permeable to propidium iodide ingress following microwave, ultrasound, and enzyme treatments. Similar to reports by Moreno-Arribas et al. [16], proline was the predominant amino acid in the wines. During secondary fermentation, free amino acids support yeast growth, and are also released during autolysis [3,30]. Preferential uptake of nitrogen compounds that can be metabolised efficiently by yeast results in depleted concentrations in wine, and subsequent accumulation of amino acids such as proline and lysine that cannot be used by yeast during anaerobic fermentation [57]. In contrast, the decrease in total free amino acid concentration between 6 and 12 months bottle age observed in the ultrasound treated wine was possibly due to reactions that breakdown amino acids (deamination) or the formation of other compounds [4,9]. The dynamics of total free amino acid content are clearly complex, being varietal and time-dependent,

given the wide range of observations from previous sparkling wine studies of increased, decreased, or static levels [4,25,37].

Although the literature is sparse on the lipid composition of sparkling wine, the effects of fatty acids and their ethyl esters on foaming properties have been reported [9,32]. We found significant interactions between treatment and bottle age with regard to the concentrations of total lipids, phospholipids, DAG, and MAG. Overall, total lipids whilst similar across treatments at age six months, progressively became more elevated at 12 months in the β-glucanase enzyme treatment wines and became more elevated again in the ultrasound and β-glucanase enzyme treatment wines at age 18 months, relative to the control and other treatments. This was driven by changes in the phospholipid and MAG components derived from the yeast plasma membrane (markers of autolysis) [23,24]. The concentration of MAG declined over time by less than half at 18 months, whereas DAG concentration increased substantially over time, more than doubling especially in the case of the enzyme treatment. We attribute this to hydrolytic activity that led to the degradation of yeast cell walls making membranes more susceptible to degradation. Prior work in our laboratory reported observations of increased numbers of cavitated yeast cells in response to the ultrasound and β-glucanase enzyme treatments relative to the control at tirage [50].

In addition to treatment-specific effects, we observed general effects of time with bottle age whereby FFA levels decreased from 7% to 1% of total lipid content and MAG decreased from 50% to 16% of total lipid content, and DAG increased from 20% to 62% of total lipid content between bottling and age 18 months. Piton et al. [32] previously reported trends of decreasing FFA and PL and increased MAG in *S. cerevisiae* cell walls during ageing in young Champagne wine. They found further decreases in PL content in yeast from late disgorged wines (19 years) and subsequent enhancement of FFA, and that MAG and DAG were converted to TAG. Changes in the lipid profile in yeast have been attributed to the destruction of yeast organelle membranes [32]. Chen et al. [31] also attributed 'yeasty' and 'fatty' sensory traits associated with beer ageing to the release of fatty acids, from induced autolysis. The presence of monoacylglycerol as the most abundant lipid class constituent in the younger sparkling wines, in this study, is noteworthy since MAG has been reported to influence foam stabilisation [58], and Pueyo et al. [29] reported that linolenic acid (free fatty acid) was the compound that best-defined foam stability.

We found no effects of yeast treatments on total phenolic concentration in the wines but significant effects of wine ageing on lees. The total phenolic content of sparkling wine increased by about 30% after 12 months bottle age compared to age six months. Furthermore, at 18 months wines contained 60% more total phenolics than at age six months. We attribute these findings to the release of phenolic compounds adsorbed by yeast during the process of autolysis with wine ageing [4] though this aspect of autolysis was not affected by the yeast treatments. In contrast to our findings, Pozo-Bayon et al. [59] reported that ageing on lees did not influence the concentration of phenolic compounds in sparkling wine possibly because very few monomeric phenolic compounds were adsorbed by yeast cells and were not released into wine with ageing. They found that differences in the low molecular weight phenolic compounds in sparkling wine were due to grape variety. Kerslake et al. [60] found that viticultural treatments e.g., pre-flowering leaf removal resulted in higher concentrations of hydroxycinnamates in Tasmanian Chardonnay base wines and increased anthocyanin concentrations in Pinot Noir base wines but the evolution of phenolic concentration following secondary fermentation and wine ageing were not reported. Esteruelas et al. [61] had reported hydroxycinnamates as the main type of phenolic compound in sparkling wine. This may explain the origin of the total phenolic contents in our tiraged Tasmanian base wine *cuvée* comprising *Vitis vinifera* L. cv. Chardonnay (63%) and Pinot Noir (37%). Furthermore, Brissonet et al. [18] found that phenolics in sparkling wine produced from Pinot Noir grapes positively affected foaming ability. Andres-Lacueva et al. [62] reported that phenolic compounds negatively affect foaming properties and caused 'gushing' during disgorging. Hence variety and blend selection, viticultural treatments and the timing of fruit harvesting may be critical to minimising the

initial abundance of phenolics. Very careful management of phenolic material is practiced in sparkling winemaking, through fining agents, so that their subsequent release during the wine ageing process does not negatively influence foaming properties. However, the ageing related increase in total phenolic concentrations is not considered a major concern because the overall concentrations are low in this study.

Similar to the phenolics, we found no effects of yeast treatments on the concentrations of total protein in wine. Exposure to the yeast treatments for longer than that used in this study may allow for further enzyme activity that may lead to differential release of proteins. Alternatively, some treatments may have been too severe and deactivated enzymes that are known to be involved in the degradation of the yeast cell wall. This might be specifically the case in the microwave treatment where the temperatures may have exceeded 50 °C [63]. Conversely, time in the bottle was a significant factor in the development of total protein concentration whereby maximum concentrations were attained after 12 months. Protein concentration has previously been positively related to sparkling wine foam height and stability [7,18,28]. The concentration of protein in wine generally decreases during ageing on lees due to protease activity and hydrolysis to lower molecular weight compounds [3]. We saw a general (non-significant) trend towards this from 12 to 18 months bottle age. This is counter to our consideration that the treatments possibly degraded the enzymes involved in the release of proteins. Also, it implies that ageing of wines on lees for 24 months would be expected to further reduce protein concentrations in the wines, thus the benefits of ageing past 18 months would need to out-weigh the potential negative impact of declining levels of proteins on foam height and stability, which in practice it seems to, given the long ageing on lees of super-premium sparkling wine available on the market.

## 5. Conclusions

This study found significant effects of bottle-ageing with important implications for sparkling winemaking. Older sparkling wines contained 60% more total phenolic substances than younger wines which are generally undesirable from the perspective of foam and flavour attributes. However, the total concentration of phenolic substances in these wines were generally low as winemakers actively manage phenolics during wine production.

In contrast, a general finding of bottle ageing was the increase in total protein concentration (which has positive implications for sparkling wine foaming properties) from 6 to 12 months bottle age. Thus, avoiding fruit with an inherently elevated abundance of phenolic substances and ensuring at least 12 months of bottle-ageing will contribute to wine quality from solely a wine-ageing perspective.

In relation to the novel treatments investigated in this study, we have demonstrated that yeast treatments added to the tirage solution are associated with the release of cell components predominantly at 18 months bottle age, relative to nil effects at six months and less distinct effects at 12 months bottle age. The microwave and β-glucanase enzyme treatments were associated with elevated total free amino acid concentration by 10%; and proline content, involved in astringency perception, by 10% in wine at 18 months bottle age. The β-glucanase enzyme and ultrasound treatments effected significant accumulation of total lipids in wine at 18 months bottle age, which was driven by the phospholipid and MAG components, whilst elevated TAG was observed in the microwave treatment at 18 months bottle age. These preliminary findings suggest that 18 months bottle age is possibly a minimum requirement for elevated concentrations of desirable compounds released into sparkling wine and that these treatments warrant further exploration as tools towards accelerated autolysis for the production of quality sparkling wines.

**Author Contributions:** Conceptualization, F.L.K. and G.B.G.; methodology, F.L.K., G.B.G., D.C.C. and S.A.S.; software, G.B.G.; validation, G.B.G., D.C.C. and F.L.K.; formal analysis, G.B.G. and S.A.S.; investigation, G.B.G. and F.L.K.; resources, G.B.G. and F.L.K.; data curation, G.B.G. writing—original draft preparation, G.B.G.; writing—review and editing, G.B.G., F.L.K., D.C.C. and S.A.S. supervision, F.L.K., D.C.C. and S.A.S.; project administration, G.B.G. and F.L.K.; funding acquisition, F.L.K. and D.C.C. All authors have read and agreed to the published version of the manuscript.

**Funding:** This research was jointly funded by Wine Australia (PhD scholarship PA001791) and the University of Tasmania, Australia (Australian Research Training Support Scholarship).

**Institutional Review Board Statement:** Ethical review and approval for this study was obtained from the Research Ethics Committee at the University of Tasmania (Ref.: H0015927).

**Data Availability Statement:** The data presented in this study are available on request from the corresponding author. The data are not publicly available due to the lead author (G.B.G.) currently undergoing PhD examination. Samples of the compounds are not available from the authors.

**Acknowledgments:** This research forms part of a larger industry-related project funded by Wine Australia Grant UT1502, led by Fiona Kerslake, from which a PhD project emerged to investigate novel methods to hasten lysis and visualise ageing on lees in sparkling winemaking. We acknowledge significant industry collaborator contributions from Tasmanian Vintners and invaluable support from collaborators at the Australian Wine Research Institute (AWRI), and Metabolomics South Australia which is funded through Bioplatforms Australia Pty Ltd. (BPA), a National Collaborative Research Infrastructure Strategy (NCRIS), and investment from the South Australian State Government and AWRI. Laboratory support from Pat Colombo and Kiera O'Brien at Tasmanian Vintners, Mitchell Gibbs and Andrew Niccum at the Sydney Institute of Marine Science (SIMS), Sydney, Australia and Sandra Garland at the University of Tasmania was very much appreciated. The contributions of Hanna Westmore, Linda Donachie and Rocco Longo at the University of Tasmania were greatly appreciated at disgorging.

**Conflicts of Interest:** The authors declare no conflict of interest. Wine Australia approved the study design (PhD scholarship PA001791).

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
