# Peer review of "Towards Accelerated Autolysis? Dynamics of Phenolics, Proteins, Amino Acids and Lipids in Response to Novel Treatments and during Ageing of Sparkling Wine"

_beverages, doi:10.3390/beverages7030050_

Round 1

Reviewer 1 Report

This manuscript contains a variety of interesting results on autolysis methods effect on sparkling wine production. This paper deserves to be published. It is well organized and results are clearly presented. Materials and methods are well described but corrections need to be done. 

Line 148 Please provide information about sulfur dioxide content

Line 156 Why this amount of treated yeast was selected? 

Line 160 Please provide information for moisture content and presence of light. Storage was performed in cube? 

Line 177 Please provide more information for enzymatic kit

Line 183 Please provide on-web accessible reference

Reviewer 2 Report

Comments to Author

Dear Sirs,

Manuscript entitled “Towards accelerated autolysis? Dynamics of phenolics, proteins, amino acids and lipids in response to novel treatments and during ageing of sparkling wine” (Manuscript ID: beverages-1281819), which was submitted to revision in Beverages adheres to the journal's standards. In my opinion it is an original and valuable research study. The presentation and discussion of the results is comprehensive, and some data are of particular interest for the industry. However, manuscript should be improved. Below I put my comments and suggestions:

Major comments

The major drawback is the statistical evaluation. I suggest you use a one-way ANOVA to compare the effect of the treatment, an analysis of variance paired-samples t-test to determine differences between the consecutive stages of winemaking, and a two-way ANOVA to determine the effect of the treatment and time and their interactions.

Minor comments

Line 10:  please change “méthode traditionelle” by “traditional method”

Line 14: please change “liqueur de tirage” by “tirage liquor”

Line 38: please change “liqueur de tirage” by “tirage liquor”

Line 44: Please change “The ageing process is influenced by the wine’s pH and the temperature and duration of wine storage” by “The ageing process is influenced by the wine’s pH, the temperature and duration of wine storage”.

Line 61: Please change “mouth feel” by “mouthfeel”

Line 92: Please change [8, 37] by [8,37]

Line 114-117: Please change by: “Palermo et al. [45] found β-glucanase enzyme-induced release of polysaccharides in two to three weeks in model wine relative to five months in conventional autolysis. Rodriguez-Nogales et al. [46,47] reported that addition of β-glucanase enzymes allowed a quicker break down of cell walls by hydrolysis of β-glucan chains releasing mannoproteins into wine.”

Line 151: Please change “liqueur de tirage” by “tirage liquor”

Line 156: Authors should have added to control wine 7.5 ml of untreated yeast culture.

Line 180-183: Are those analyzes published in the article? If not, remove the analysis from the Analysis of basic wine composition section.

Line 188: Please change “Haze-forming proteins ,” by “Haze-forming proteins,”

Line 194: Replace “was 0.1 % TFA/ACN)” with “was 0.1 % TFA/ACN”

Line 208-217: rewrite, it is not clear.

Line 256: “struments CO. Ltd.) Lipid” change by “struments CO. Ltd.). Lipid”

Line 355: Replace “(1.04 μg mL-1)” with “(1.14 μg mL-1)”

Table 1,2, and 3: write equal footnotes

Line 390: Please, add “At 18 months of aging, the total free…”

In the References section, the writing manner of some references did not follow the style of this journal and there are some typographical errors. Authors must check and revise these errors carefully. For example: ref 18, ref 44, ref 47, ref 49, ref 63, etc.

Round 2

Reviewer 2 Report

The manuscript has been sufficiently improved to warrant publication in Beverages